# Temporal Convolutional Networks Applied to Energy-Related Time Series Forecasting

**Pedro Lara-Benítez** *,† , **Manuel Carranza-García** † , **José M. Luna-Romera** and **José C. Riquelme**

Division of Computer Science, University of Sevilla, ES-41012 Seville, Spain;
mcarranzag@us.es (M.C.-G.); jmluna@us.es (J.M.L.-R.); riquelme@us.es (J.C.R.)
* Correspondence: plbenitez@us.es
† These authors contributed equally to this work.

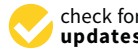

**Featured Application: Energy demand forecasting to improve power generation management.**

**Abstract:** Modern energy systems collect high volumes of data that can provide valuable information about energy consumption. Electric companies can now use historical data to make informed decisions on energy production by forecasting the expected demand. Many deep learning models have been proposed to deal with these types of time series forecasting problems. Deep neural networks, such as recurrent or convolutional, can automatically capture complex patterns in time series data and provide accurate predictions. In particular, Temporal Convolutional Networks (TCN) are a specialised architecture that has advantages over recurrent networks for forecasting tasks. TCNs are able to extract long-term patterns using dilated causal convolutions and residual blocks, and can also be more efficient in terms of computation time. In this work, we propose a TCN-based deep learning model to improve the predictive performance in energy demand forecasting. Two energy-related time series with data from Spain have been studied: the national electric demand and the power demand at charging stations for electric vehicles. An extensive experimental study has been conducted, involving more than 1900 models with different architectures and parametrisations. The TCN proposal outperforms the forecasting accuracy of Long Short-Term Memory (LSTM) recurrent networks, which are considered the state-of-the-art in the field.

**Keywords:** deep learning; energy demand; temporal convolutional network; time series forecasting

## 1. Introduction

Forecasting electricity demand is currently amongst the most important challenges for the industries. Due to the increasingly high level of electricity consumption, electrical companies need to efficiently manage the production of energy. Sustainable production plans are required to meet demands and account for important challenges of this century such as global warming and the energy crisis. Smart meters now provide useful data that can help to understand consumption patterns and monitor power demand more efficiently. Data mining techniques can use this information to learn from historical past data and predict the expected demand to make decisions accordingly. Obtaining accurate forecasts can be essential for the future electricity market considering the increasing penetration of renewable energies. However, forecasting power demand is a complex task that involves many factors and requires sophisticated machine learning models to produce high-quality predictions.

Statistical-based models, such as the Box–Jenkins model called ARIMA, were for many years the state-of-the-art for electricity time series forecasting [1,2]. However, machine learning models have proven to provide better performance for problems of this domain. Artificial neural networks

(ANNs) [3], support vector machines (SVMs) [4,5], and regression trees [6] have been applied successfully for diverse power demand prediction tasks. More recently, deep learning (DL) has emerged as a very powerful approach for time series forecasting. DL models are especially suitable for big-data temporal sequences due to their capacity to extract complex patterns automatically without feature extraction preprocessing steps [7]. As an evolution from simple ANNs, deep, fully connected networks have been applied for load forecasting problems [8]. However, fully connected networks are unable to capture the temporal dependencies of a time series. Consequently, more specialised DL models such as recurrent neural networks (RNNs) and convolutional neural networks (CNNs) started to gain importance in the time series forecasting field. These networks can efficiently encode the underlying patterns of time series by transforming the temporal problem into a spatial architecture [9].

In the recent literature, a significant number of studies presenting results of the application of RNNs to energy-related time series forecasting can be found [10,11]. Among all existing RNN architectures, long short-term memory (LSTM) networks have been the most popular due to their capacity to solve problems of previous RNN such as gradient explosion and vanishing gradient [12]. It has been considered a standard forecasting model for several tasks such as traffic prediction [13], solar power forecasting [14], financial market predictions [15], and electricity price prediction [16]. Although CNNs were originally designed for computer vision tasks, they are also suitable for time series data since they can extract high-level features from data with a grid topology. Despite the popularity of RNNs, several works using convolutional networks can be found. In both [17,18], the authors proposed CNN models for short-term load forecasting that provides comparable results to LSTM models. Other works have been able to build deep convolutional networks that can outperform LSTM networks for electricity demand [19] and solar power data problems [20]. Furthermore, in all these works, the CNN models proved to be more suitable for real-time applications given their faster training and testing execution time. The properties of local connectivity and parameter sharing of convolutional networks reduce the number of trainable parameters compared to RNNs, hence they can be trained more efficiently. There have also been proposals using hybrid models that combine convolutional and LSTM layers. In [21], the output feature maps of a CNN are fed to a RNN that provides the prediction. Other approaches consider combining the features extracted in parallel from a CNN and a LSTM to improve the forecasting using electricity demand data [22] or financial data [23]. These ensemble proposals can enhance the predictive performance by fusing the long-term patterns captured by the LSTM and the local trend features obtained with the CNN.

More recently, a specialised CNN architecture known as temporal convolutional networks (TCN) has acquired popularity due to their suitability to deal with time series data. TCNs were first proposed in [24], in which they were compared to several RNNs over sequence modelling tasks. TCNs use causal dilated causal convolution in order to be able to capture longer-term dependencies and prevent information loss. Furthermore, they present other advantages over RNNs such as lower memory requirements, parallel processing of long sequences as opposed to the sequential approach of RNNs, and a more stable training scheme. Several works have already successfully used TCNs for time series forecasting tasks: the original architecture using stacked dilated convolutions was proposed in [25] to improve the performance of LSTM networks for financial domain problems; Ref. [26] designed a deep TCN for multiple related time series with an encoder–decoder scheme, evaluating over data from the sales domain; the study in [27] proposed a multivariate time series forecasting model for meteorological data, which outperformed several popular deep learning models. However, to the best of our knowledge, the potential of TCNs has not yet been explored for univariate time series forecasting problems related to electricity demand data.

In this work, we study the applicability and performance of TCNs for multistep time series forecasting over two energy-related datasets. With the first dataset, we build a deep learning model to forecast the electricity demand in Spain based on the historical consumption data over five years. In the second dataset, the problem is to forecast the expected energy consumption of charging stations for electric vehicles in Spain. Our aim in this study is to present a deep learning model that uses a TCN



to obtain high accuracy on time series forecasting. We present the results obtained with several TCN architectures and perform an extensive comparison with different LSTM models, which has been so far the most extended approach for these types of problems. In the experimental study, we carry out an extensive parameter search process which involves 1998 different network architectures.

In summary, the main scientific contributions of this paper can be condensed as follows:

- A temporal convolutional neural network model to achieve high accuracy in forecasting over energy demand time series;
- A thorough experimental study, comparing the performance of temporal convolutional with long short-term memory networks for time series forecasting.

The rest of the paper is organised as follows: Section 2 describes the materials used, the methodology, and the experiments carried out; in Section 3, the experimental results obtained are reported and discussed; Section 4 presents the conclusions and future work.

## 2. Materials and Methods

In this section, we present the datasets selected for the study, the methodology to perform time series forecasting using deep learning models, and the details of the experimental study carried out.

### 2.1. Datasets

In this subsection, we present the two energy-related datasets selected for the study.

#### 2.1.1. Electric Demand in Spain

The first dataset used in the experimental study covers the national electrical energy demand in Spain ranging from 2014-01-02T00:00 to 2019-11-01T23:50. During this period, the consumption was measured every ten minutes which makes a time series with a total length of 306,721 measurements. The data was provided by Red Eléctrica de España (the Spanish public grid) and is publicly available at [28]. The dataset was divided into two sets: the training set contains 245,376 samples corresponding to the period from 2014-01-02T00:00 to 2018-09-02T00:50; and the test set contains 61,343 samples comprising the period from 2018-09-02T01:00 to 2019-11-01T23:40. As it has been done in previous studies using this data [29], we defined the forecasting horizon to be 4 hours, which involves a prediction of 24 time-steps.

Figure 1 plots the electric demand data at different scales. As can be seen, the time series presents both weekly and daily seasonality. In general, the demand is higher at weekdays than at weekends and suffers a severe drop during night-time every day.

#### 2.1.2. Electric Vehicles Power Consumption

The second dataset gathers information about power consumption in charging stations for electric vehicles (EV) in Spain. This data was also obtained from Red Eléctrica de España and can be found at [30]. In the near future, governments will have to build infrastructures that can fulfil the demands of the increasing EV fleet. Given their limited autonomy, the prediction of EV consumption seems crucial to efficiently manage the power supply. In this article, we followed the same steps to generate the EV demand time series as in [31]. The data was collected hourly and ranges from 2015-03-02T00:00 to 2016-05-31T23:00. For each geographical area, we obtained a single value of power consumption for every hour. Later, in order to obtain a single time series, the different zones were aggregated giving the total energy consumption of the Spanish EV. The forecasting horizon was set to be 48 h, which involves a prediction of 48 time-steps. The dataset is divided into two sets: the training set contains 8759 samples corresponding to the period from 2015-03-02T00:00 to 2016-02-29T23:00; and the test set contains 2207 samples comprising the period from 2016-03-01T00:00 to 2016-05-31T23:00.

As can be seen in Figure 2, the time series presents weekly and daily patterns. Since electric vehicles are most commonly charged at night, the time series presents peak values of demand during

the first six hours of each day. Only Sundays and Mondays present a slightly different pattern, as it is displayed in Figure 2c.

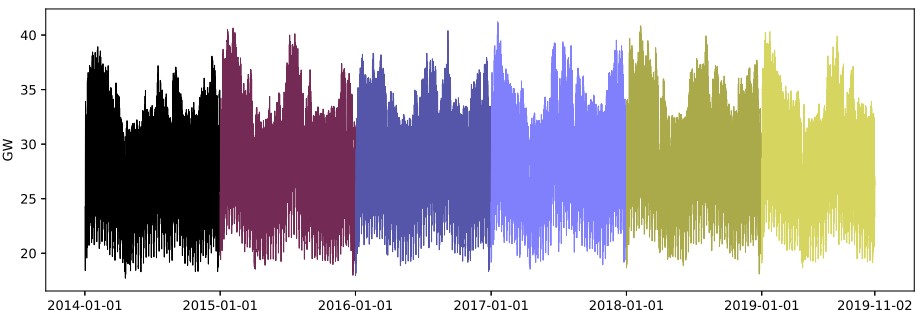

(**a**) Complete time series showing the evolution of the electric demand in Spain from 2014 to 2019.

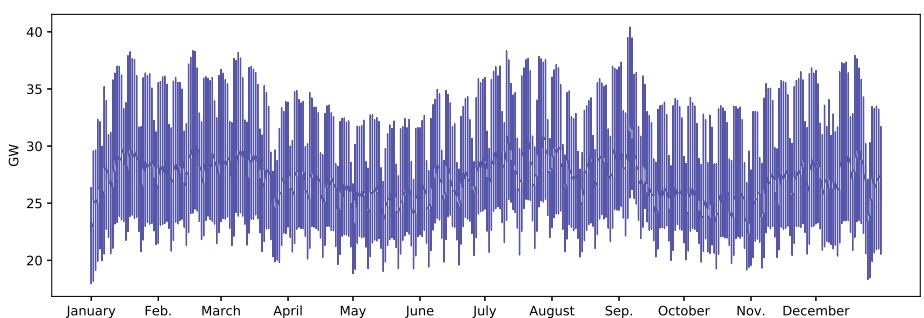

(**b**) Evolution of the electric demand in Spain during 2016.

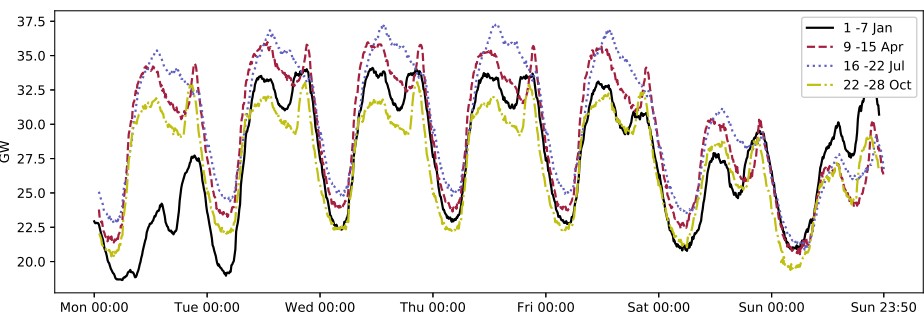

(**c**) Evolution of the electric demand during four different weeks of 2018.

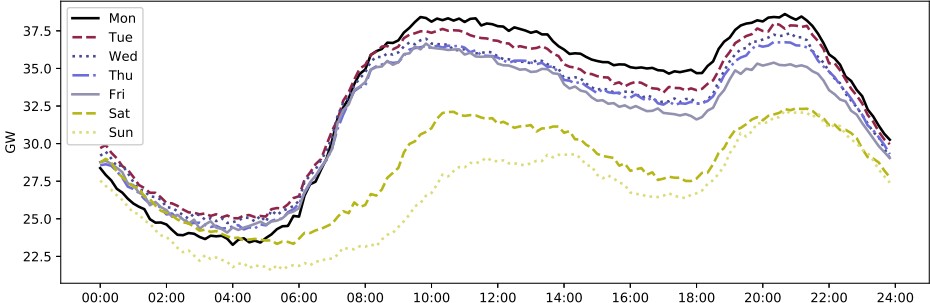

(**d**) Evolution of the electric demand within each day of the first week of February 2019.

**Figure 1.** Line plots illustrating the electric demand time series data at different scales.

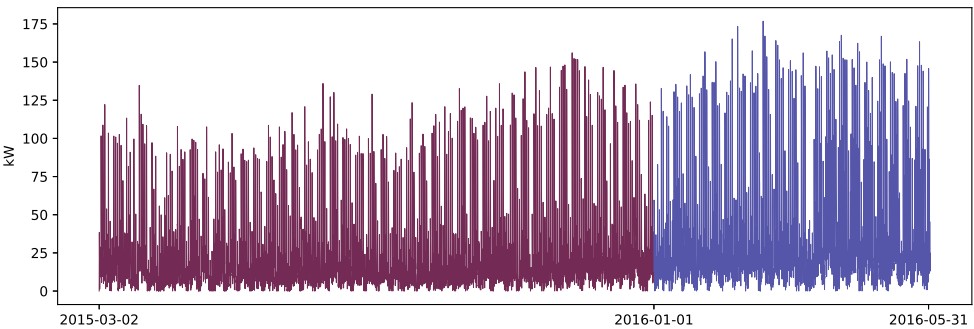

(**a**) Complete time series showing the evolution of electric vehicle (EV) power consumption from March 2015 to end of May 2016.

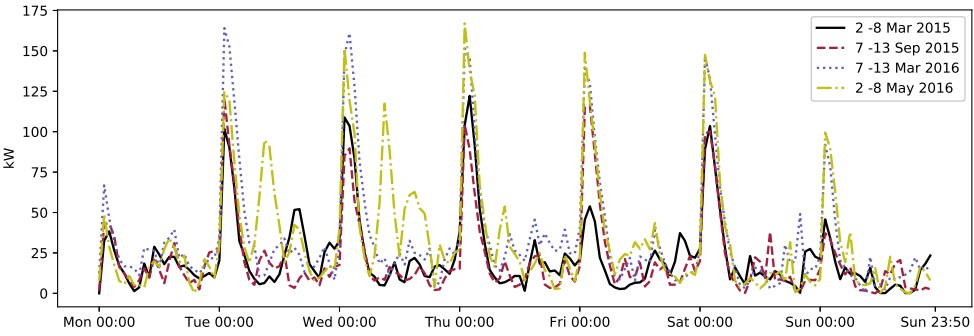

(**b**) Evolution of the EV power consumption during four different weeks.

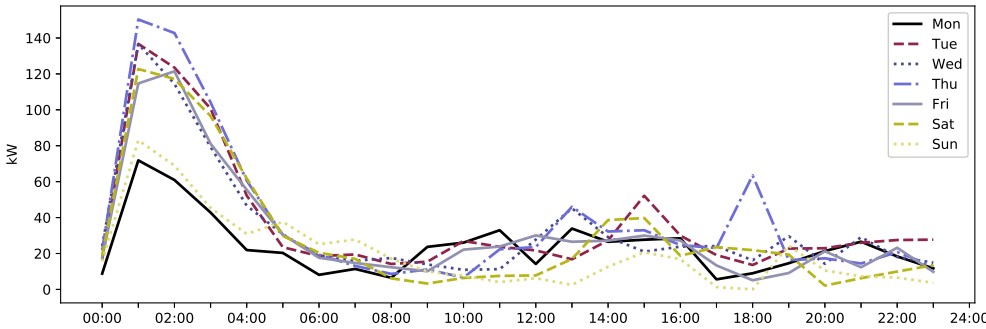

(**c**) Evolution of the EV power consumption within each day of the first week of February 2016.

**Figure 2.** Line plots illustrating the electric vehicle power consumption data at different scales.

## 2.2. Methodology

In this subsection, we describe the required data preprocessing steps and the fundamental concepts behind temporal convolutional networks.

### 2.2.1. Data Preprocessing

In order to train a deep learning model that can predict several time-steps, a preprocessing stage is needed to transform the original time series data. First, we perform min-max normalisation to the entire sequence to scale the values between 0 and 1, which helps to improve the convergence of deep networks. Secondly, we transform the sequence into instances that can be used to feed the

network. There exist several strategies to deal with multistep forecasting problems [32]: the recursive strategy, which performs one-step predictions and feeds the result as the last input for the next prediction; the direct strategy, which builds one model for each time step; and the multi-output approach, which outputs the complete forecasting horizon vector using just one model. As suggested in recent forecasting studies that use neural networks [33,34], in this work, we adopt the MIMO strategy (Multi-Input Multi-Output) which belongs to the last category. Instead of forecasting each time-step independently, the MIMO approach can model the dependencies between the predicted values since it outputs the complete forecasting window. Furthermore, this strategy avoids the accumulated errors over predictions that appear in the recursive strategy.

Following this approach, a moving window scheme is used to create the input–output pairs that will be fed to the neural network. All deep learning models used in this study accept a fixed-length window as input and have an output dense layer with as many neurons as the forecasting horizon defined for each problem (24 for electricity demand and 48 for electric vehicle demand). Figure 3 illustrates the process of applying the moving window over the complete time series. As can be seen, the window slides and obtains an input–output instance at each position. While the output window size is defined by the problem, the input window size has to be decided. The optimal value can be different depending on the data, the designed model, and the forecasting horizon. In our study, we have experimented with three different sizes for the input window of each problem. The values have been carefully selected, considering the characteristics and seasonality of the datasets. For the electricity demand, we evaluate using 144, 168, and 268 time-steps as input window (which corresponds to 24, 28, and 48 h, respectively). For the power demand of electric vehicles, we consider 168, 336, and 672 time-steps as input window (which corresponds to 7, 14, and 28 days, respectively).

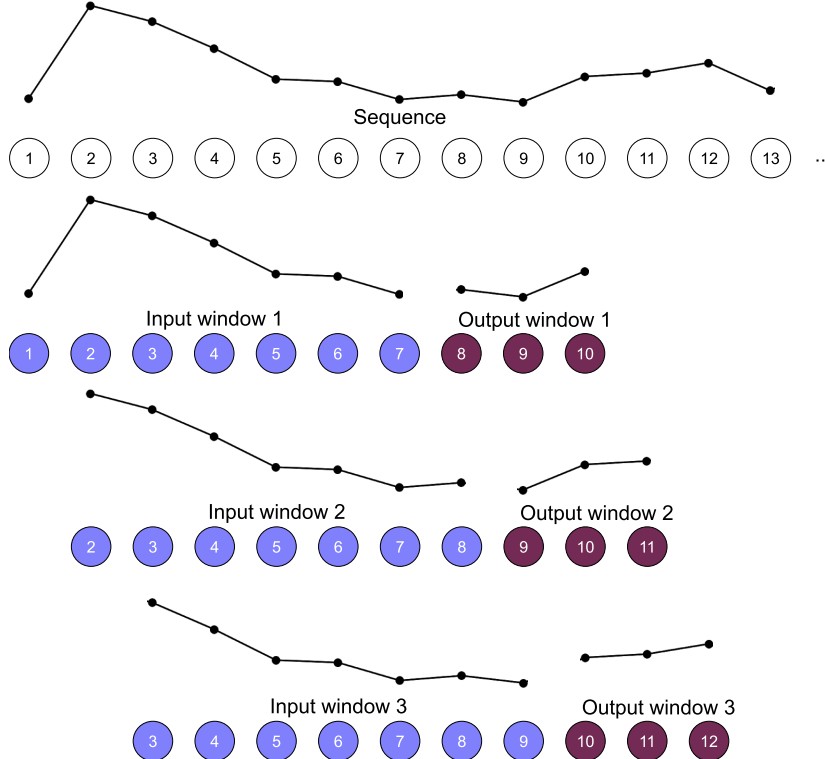

**Figure 3.** Moving window procedure that obtains the input–output instances. In this example, the input and output windows have lengths of 7 and 3, respectively.

### 2.2.2. Temporal Convolutional Neural Network

TCNs are a type of convolutional neural network with a specific design that makes them suitable for handling time series. TCNs satisfy two main principles: the network's output has the same length as the input sequence (similarly to LSTM networks); and they prevent leakage of information from future to the past by using causal convolutions [24]. Causal convolution differs from standard convolution in the fact that the convolutional operation performed to obtain the output at time $t$ does not take future values as inputs. This implies that, using a kernel size $k$, the output $O_t$ is obtained using the values of $X_{t-(k-1)}, X_{t-(k-2)}, \ldots, X_{t-1}, X_t$ (Figure 4). Zero-padding of length $k-1$ is used at every layer to maintain the same length as the input sequence.

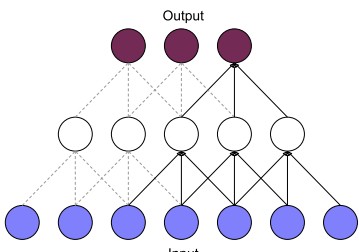
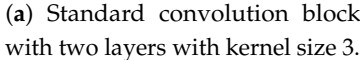
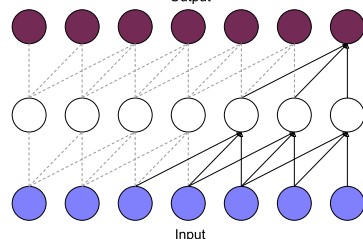
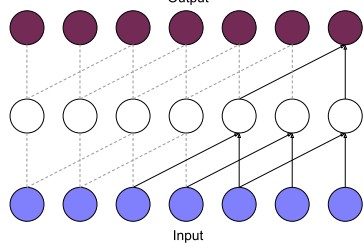

(**a**) Standard convolution block with two layers with kernel size 3.

(**b**) Causal convolution block with two layers with kernel size 3.

(**c**) Dilated causal convolution block with two layers with kernel size 2, dilation rate 2.

**Figure 4.** Differences between (**a**) standard convolutional network, (**b**) causal convolutional network, and (**c**) dilated causal convolutional network.

Furthermore, with the aim of capturing longer-term patterns, TCNs use one-dimensional dilated convolutions. This convolution increases the receptive field of the network without using pooling operations, hence there is no loss of resolution [35]. Dilation consists of skipping $d$ values between the inputs of the convolutional operation, as can be seen in Figure 4c. The complete dilated causal convolution operation over consecutive layers can be formulated as follows [36]:

$$x_l^t = g\left(\sum_{k=0}^{K-1} w_l^k \, x_{(l-1)}^{(t-(k\times d))} + b_l\right),\qquad(1)$$

where $x_l^t$ is the output of the neuron at position $(t)$ in the $l$-th layer; $K$ is the width of the convolutional kernel; $w_l^k$ stands for the weight of position $(k)$; $d$ is the dilation factor of the convolution; and $b_l$ is the bias term. Rectified Linear Units (ReLU) layers are used as activation function ($g(x) = max(0, x)$) [37]. Another common approach to further increase the network's receptive field is to concatenate several TCN blocks, as can be seen in Figure 5 [38]. However, this leads to deeper architectures with many more parameters which complicates the learning procedure. For this reason, a residual connection is added to the output of each TCN block. Residual connections were proposed by [39] in order to improve performance in very deep architectures, and consist of adding the input of a TCN block to its output ($o = g(x + F(x))$).

All these characteristics make TCNs a very suitable deep learning architecture for complex time series problems. The main advantage of TCNs is that, similarly to RNNs, they can handle variable-length inputs by sliding the one-dimensional causal convolutional kernel. Furthermore, TCNs are more memory efficient than recurrent networks due to the shared convolution architecture which allows them to process long sequences in parallel. In RNNs, the input sequences are processed sequentially, which results in higher computation time. Moreover, TCNs are trained with the standard backpropagation algorithm, hence avoiding the gradient problems of the backpropagation-through-time algorithm (BPTT) used in RNN [40].

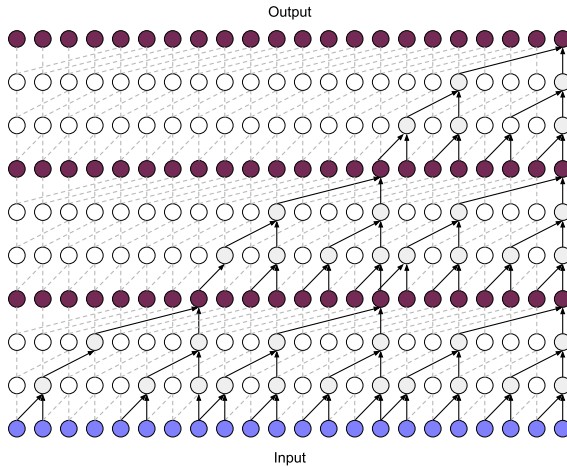

**Figure 5.** Temporal Convolutional Networks (TCN) model with 3 stacked blocks. Each block has 3 convolutional layers with kernel size 2 and dilations [1, 2, 4].

### 2.3. Experimental Study

In this subsection, we present the design of the experimental study carried out over the two energy-related datasets. Furthermore, we also describe the details of the parameter search process for each model architecture.

#### 2.3.1. Models

The aim of this work is to build a TCN-based deep learning model to improve the performance in energy demand forecasting tasks, in terms of both accuracy and efficiency. In order to compare the effectiveness of TCNs for this problem, we also evaluate the performance of recurrent LSTM networks—that have so far been considered the state-of-the-art in forecasting. However, given the high complexity of deep learning models, finding optimal values for the hyperparameters of these networks is a very challenging task. Therefore, we have conducted an extensive experimental study that involves more than 1900 combinations of parameters that build different convolutional and recurrent architectures. An important hyperparameter that is common to both types of architectures is the size of the input window. The possible values for past history were defined in Section 2.1 and depend on the data characteristics and seasonality. Additionally, we have searched for the best values of several parameters that are specific for TCN or LSTM architectures. In the case of TCNs, we have experimented with a different number of filters and stacked residual blocks, kernel sizes, and dilations factors. In the case of LSTMs, we have experimented with a different number of stacked layers and units. Furthermore, we have also studied the effect of training parameters in the performance of all models, such as the batch size and the number of epochs. The Adam optimiser has been selected for training the models, which has an adaptive learning rate that can improve the convergence speed of deep networks [41]. The mean absolute error (MAE) has been used as the loss function for all experiments.

Table 1 displays all TCN architecture configurations that have been tested over both datasets. The parameter search process has been designed considering the receptive field of neurons inside the network, that can be calculated as follows: (*receptive field = no. stacked blocks × kernel size × last dilation factor*). Depending on the length of the past history window, we carefully select possible values for kernel size, stacked blocks, and dilations so that the receptive field covers the whole input sequence. For instance, if the number of stacked block increases, less dilated convolutional layers are needed, as can be seen in Table 1a,b. All these architectures are then tested with all combinations of parameters displayed in Table 1c that are common for both datasets (number of convolutional filters, epochs, and batch size). Overall, 756 (28 models from Figure 7a × 3 numbers of filters × 3 batch size ×

3 number of epochs from Table 1c) experiments with different configurations using TCNs have been conducted for the electric demand dataset, and 513 (12 from Figure 7b $\times$ 3$\times$3$\times$3 from Table 1c) for the EV power consumption data.

**Table 1.** Architecture configuration of all TCN models for each dataset and common parameter search.

**a** TCN architectures depending on the past history for electric demand data.

| Past History | TCN Model Architecture | | |
| --- | --- | --- | --- |
| | Kernel Size | No. Blocks | Dilations |
| 144 | 2 | 3 | [1, 3, 6, 12, 24] |
| | 3 | 1 | [1, 3, 6, 12, 24, 48] |
| | | 2 | [1, 3, 6, 12, 24] |
| | | 3 | [1, 2, 4, 8, 16] |
| | | 4 | [1, 3, 6, 12] |
| | 4 | 3 | [1, 3, 6, 12] |
| | | 4 | [1, 3, 9] |
| | 6 | 1 | [1, 3, 6, 12, 24] |
| | | 2 | [1, 3, 6, 12] |
| | | 3 | [1, 2, 4, 8] |
| | | 4 | [1, 3, 6] |
| 168 | 2 | 3 | [1, 5, 7, 14, 28] |
| | 3 | 1 | [1, 5, 7, 14, 28, 56] |
| | | 2 | [1, 5, 7, 14, 28] |
| | | 4 | [1, 5, 7, 14] |
| | 4 | 3 | [1, 5, 7, 14] |
| | 6 | 1 | [1, 5, 7, 14, 28] |
| | | 2 | [1, 5, 7, 14] |
| | | 4 | [1, 5, 7] |
| 288 | 2 | 3 | [1, 3, 6, 12, 24, 48] |
| | 3 | 2 | [1, 3, 6, 12, 24, 48] |
| | | 3 | [1, 2, 4, 8, 16, 32] |
| | | 4 | [1, 3, 6, 12, 24] |
| | 4 | 3 | [1, 3, 6, 12, 24] |
| | 6 | 1 | [1, 3, 6, 12, 24, 48] |
| | | 2 | [1, 3, 6, 12, 24] |
| | | 3 | [1, 2, 4, 8, 16] |
| | | 4 | [1, 3, 6, 12] |

**b** TCN architectures depending on the past history for EV power consumption data.

| Past History | TCN Model Architecture | | |
| --- | --- | --- | --- |
| | Kernel Size | No. Blocks | Dilations |
| 168 | 2 | 3 | [1, 5, 7, 14, 28] |
| | 3 | 1 | [1, 5, 7, 14, 28, 56] |
| | | 2 | [1, 5, 7, 14, 28] |
| | | 3 | [1, 5, 7, 14] |
| | 4 | 3 | [1, 5, 7, 14] |
| | 6 | 1 | [1, 5, 7, 14, 28] |
| | | 2 | [1, 5, 7, 14] |
| | | 4 | [1, 5, 7] |
| 336 | 2 | 3 | [1, 5, 7, 14, 28, 56] |
| | 3 | 2 | [1, 5, 7, 14, 28, 56] |
| | | 4 | [1, 5, 7, 14, 28] |
| | 4 | 3 | [1, 5, 7, 14, 28] |
| | 6 | 1 | [1, 5, 7, 14, 28, 56] |
| | | 2 | [1, 5, 7, 14, 28] |
| | | 4 | [1, 5, 7, 14] |
| 672 | 3 | 4 | [1, 5, 7, 14, 28, 56] |
| | 4 | 3 | [1, 5, 7, 14, 28, 56] |
| | 6 | 2 | [1, 5, 7, 14, 28, 56] |
| | | 4 | [1, 5, 7, 14, 28] |

**c** Parameter search for TCN models.

| TCN Parameters | |
| --- | --- |
| **No. of filters** | {32, 64, 128} |
| **Batch size** | {64, 128, 255} |
| **No. of epochs** | {25, 50, 100} |

Table 2 presents the parameters that have been studied for LSTM networks. Given the sequential processing nature of these networks, they can effectively cover the complete input sequence and capture long-term dependencies. Therefore, the parameter search, in this case, is based on trying different combinations of LSTM units that can process sequences and feed the output to subsequent stacked layers. We also consider the same possible values as above for the input window length. A total of 243 (3 past history $\times$ 3$\times$3$\times$3 from Table 2) experiments with different LSTM models have been carried out for each dataset.

**Table 2.** Parameter search for LSTM models for both datasets.

| LSTM Parameters | | |
|---|---|---|
| No. stacked layers | | {1, 2, 3} |
| LSTM Units | | {32, 64, 128} |
| Training parameters | Batch size | {64, 128, 256} |
| | No. of epochs | {25, 50, 100} |

2.3.2. Evaluation Metric

For evaluating the predictive performance of all models we use the weighted absolute percentage error (WAPE). This metric has been suggested by recent studies dealing with energy demand data [31]. Electric industries are interested in knowing the deviation in watts for better load generation planning. Therefore, WAPE is very suitable for this context since it provides absolute error values. WAPE can be defined as follows:

$$WAPE(y, o) = \frac{MAE(y, o)}{mean(y)} = \frac{mean(|y - o|)}{mean(y)},$$  (2)

where $y$ and $o$ are two vectors with the real and predicted values, respectively, that have a length equal to the forecasting horizon.

## 3. Results and Discussion

This section reports and discusses the results obtained from the experiments carried out with the different model architectures presented in the previous section. For all tests, we have used a computer with an Intel Core i7-770K CPU and a NVIDIA GeForce GTX 1080 8GB GPU. The source code and the complete experimental results report can be found at [42].

*3.1. Forecasting Accuracy*

Figure 6 shows a comparison between the overall performance of TCN and LSTM models. It presents the distribution of the results obtained for each dataset with all architectures depending on the past history window length. In general, it can be seen that TCN models achieve a better predictive accuracy compared to LSTM networks. In almost all cases, groups of different TCN architectures with the same input window have a smaller deviation. This implies that TCN models are less sensitive to the hyperparameter selection as long as the past history remains fixed. The most robust performance is given by TCN architectures using input windows of 288 for the electric demand data and 168 for the EV power demand data. Only in the case of the EV power consumption using a very long past history (672 h) do TCN models struggle to get a stable performance. Using longer input windows implies more trainable parameters and can complicate the learning procedure of very deep convolutional networks. With respect to the LSTM models, the worst results are obtained when using the largest history size. This suggests that the proposed recurrent networks are not able to efficiently process long input sequences, and can better capture temporal dependencies using smaller windows. The difference in performance between TCN and LSTM is higher in the electric demand dataset, which is also the longest time series. This indicates that the acquisition of high volumes of data is a fundamental step in order to obtain robust TCN-based deep learning models.

Table 3 presents the TCN and LSTM architecture that obtained the best WAPE result for each past history value. The highest accuracy for both datasets has been obtained with very similar TCN architectures. In the case of the electric demand data, the best result (0.0093 WAPE) has been obtained using 48 h (288 time-steps) as past history. This TCN architecture, which is represented in Figure 7a, has two residual blocks with five convolutional layers of kernel size 6, 128 filters, and dilations of 1, 3, 6, 12, and 24. This model has been trained over 50 epochs with a batch size of 128 instances. In contrast, the best LSTM model (0.0105 WAPE) for this dataset uses the smallest possible input

window, which is 24 h. This LSTM model consists of 2 stacked layers with 128 units, and was trained over 50 epochs using 64 as batch size. In general, for this dataset, it can be seen that the best results have been provided by stacking two or three layers that use the greatest amount of filters or units (128). For TCNs, the highest kernel size (6) has proved to be the most effective for capturing local trend patterns in this long time series with daily and weekly seasonality. Furthermore, TCN blocks needed at least four convolutional layers with increasing dilation factors to achieve the highest accuracy. Both types of network achieve better results when training for 50 epochs, which suggest that an excessive number of iterations may cause overfitting issues. Concerning the batch size, the optimal value is 128 for almost all cases. Deep networks lose generalisation capacity when trained using large batches since they often converge to sharp minimisers [43], hence choosing smaller values can be beneficial.

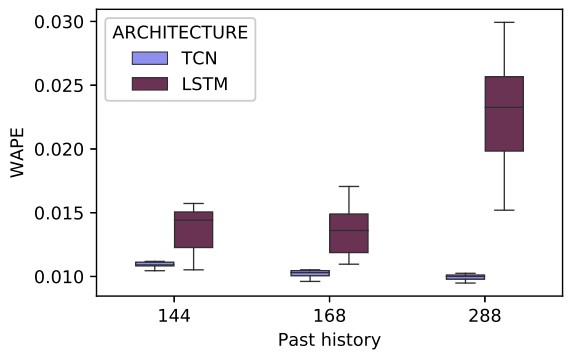

(**a**) Results for the electric demand data.

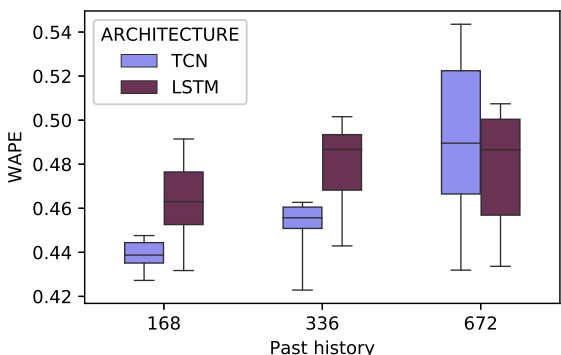

(**b**) Results for the EV power consumption data.

**Figure 6.** Distribution of weighted absolute percentage error (WAPE) results for all architectures depending on the past history window.

**Table 3.** WAPE results of the best TCN and Long Short-Term Memory (LSTM) models for each possible value of input window.

**a** Best results for electric demand data.

| Past History | | Model Description | | | | | | | | WAPE |
|---|---|---|---|---|---|---|---|---|---|---|
| Time-Steps | Hours | # | Type | Blocks | Filters | Kernel | Dilations | Batch Size | Epochs | |
| 144 | 24 | 1 | LSTM | 3 | 128 | | | 128 | 50 | 0.0105 |
| | | 2 | TCN | 2 | 128 | 6 | [1, 3, 6, 12] | 256 | 50 | 0.0104 |
| 168 | 28 | 3 | LSTM | 3 | 128 | | | 128 | 50 | 0.0109 |
| | | 7 | TCN | 2 | 128 | 6 | [1, 5, 7, 14] | 128 | 50 | 0.0096 |
| 288 | 48 | 5 | LSTM | 3 | 128 | | | 128 | 50 | 0.0151 |
| | | 6 | TCN | 2 | 128 | 6 | [1, 3, 6, 12, 24] | 128 | 50 | **0.0093** |

**b** Best results for the EV power consumption data.

| Past History | | Model Description | | | | | | | | WAPE |
|---|---|---|---|---|---|---|---|---|---|---|
| Time-Steps | Days | # | Type | Blocks | Filters | Kernel | Dilations | Batch Size | Epochs | |
| 168 | 7 | 7 | LSTM | 2 | 128 | | | 128 | 50 | 0.4317 |
| | | 8 | TCN | 1 | 128 | 3 | [1, 5, 7, 14, 28, 56] | 128 | 50 | 0.4272 |
| 336 | 14 | 9 | LSTM | 3 | 64 | | | 64 | 50 | 0.4429 |
| | | 10 | TCN | 2 | 128 | 3 | [1, 5, 7, 14, 28, 56] | 64 | 100 | **0.4228** |
| 672 | 28 | 11 | LSTM | 2 | 64 | | | 64 | 50 | 0.4336 |
| | | 12 | TCN | 2 | 64 | 6 | [1, 5, 7, 14, 28, 56] | 64 | 50 | 0.4319 |

For the electric vehicle power consumption data, the best result (0.4228 WAPE) has been obtained with a past history of 14 days (336 time-steps). This TCN architecture, which is represented in Figure 7b, has two residual blocks with six convolutional layers of kernel size 3, 128 filters, and dilations of 1, 5, 7,

14, 28, and 56. This model has been trained over 100 epochs with a batch size of 64 instances. Similar to the previous dataset, the best LSTM model (0.04317 WAPE) uses the smallest possible input window, which is 7 days. This LSTM model consists of 2 stacked layers with 128 units and was trained over 50 epochs using 128 as batch size. Given the different nature of this dataset, the best configurations of architectures present several differences. For TCNs, a smaller kernel was able to extract the underlying patterns more accurately. Furthermore, a smaller batch size was better in almost all cases since the EV time series has fewer instances to train with.

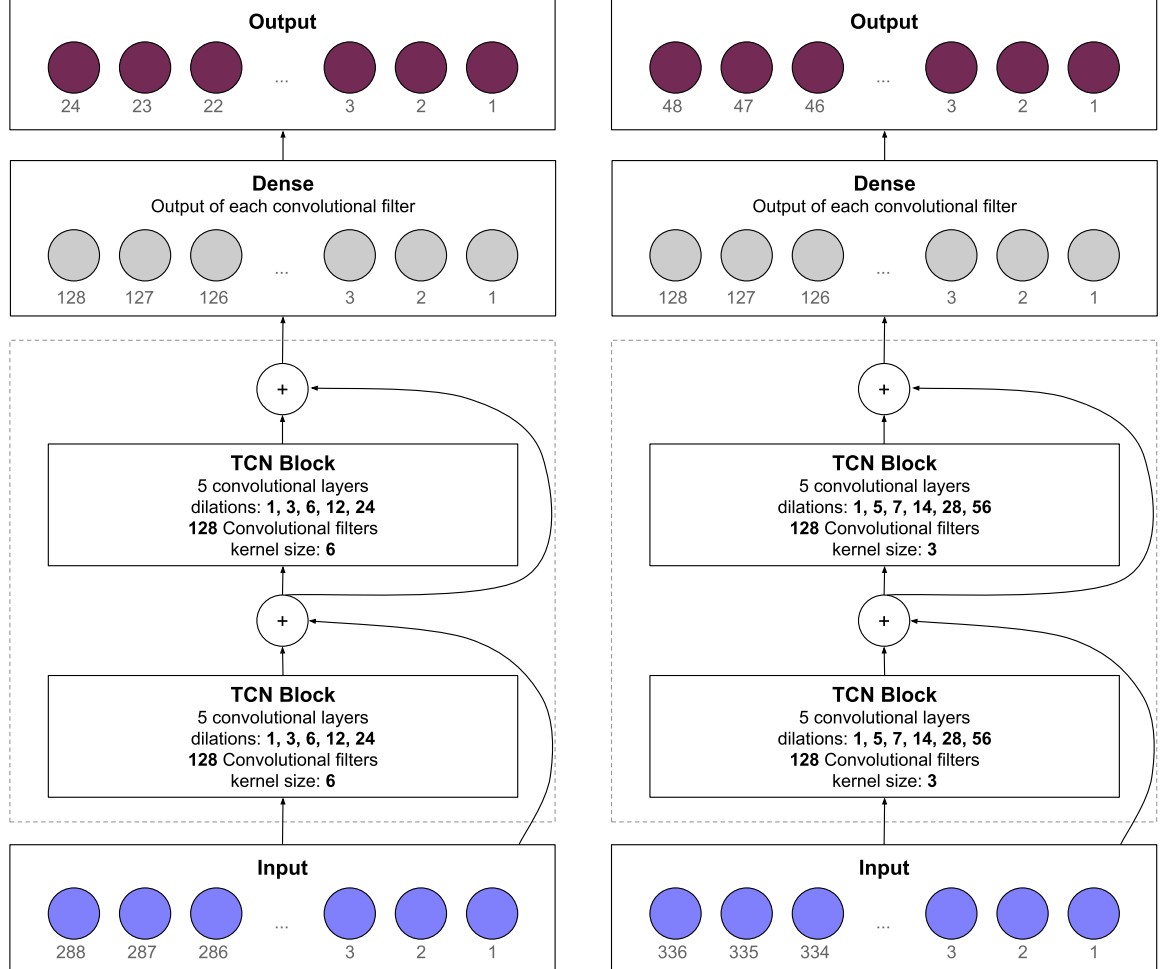

(**a**) Best TCN architecture for the electric demand data.     (**b**) Best TCN architecture for the EV data.

**Figure 7.** Architecture of the best TCN model for each dataset.

In addition, Figure 8 shows the evolution of the training and validation loss for the best TCN and LSTM models. These plots can help to compare the learning process of the different architectures presented in Table 3. It can be seen that TCNs have a more stable loss optimisation procedure compared to the LSTM models, which can be associated with the use of the standard backpropagation method. The training curves of convolutional networks suffer fewer oscillations than the recurrent approach. For the electric demand dataset, TCN models converge faster to a lower validation loss. However, for the EV power demand data, LSTMs have an inferior validation loss at the initialisation point and hence converge more rapidly.

In general, the obtained results demonstrate that TCN models can outperform the forecasting accuracy of LSTM models. TCNs have shown a more reliable performance regardless of the selected architecture and parametrisation. The dilated casual convolutions employed by TCNs were better at capturing long-term dependencies than recurrent units. Other important conclusions can be drawn

from the experimental results of this study. As can be seen, the best LSTM models for both datasets use the smallest possible input window. This indicates that the sequential processing of recurrent networks is not optimal for dealing with very long input sequences. In contrast, the use of residual connections in TCNs allows to increase the depth of the network and effectively encode longer sequences. Furthermore, another aspect to consider is that the best performance was obtained when stacked layers used as many filters or units as possible.

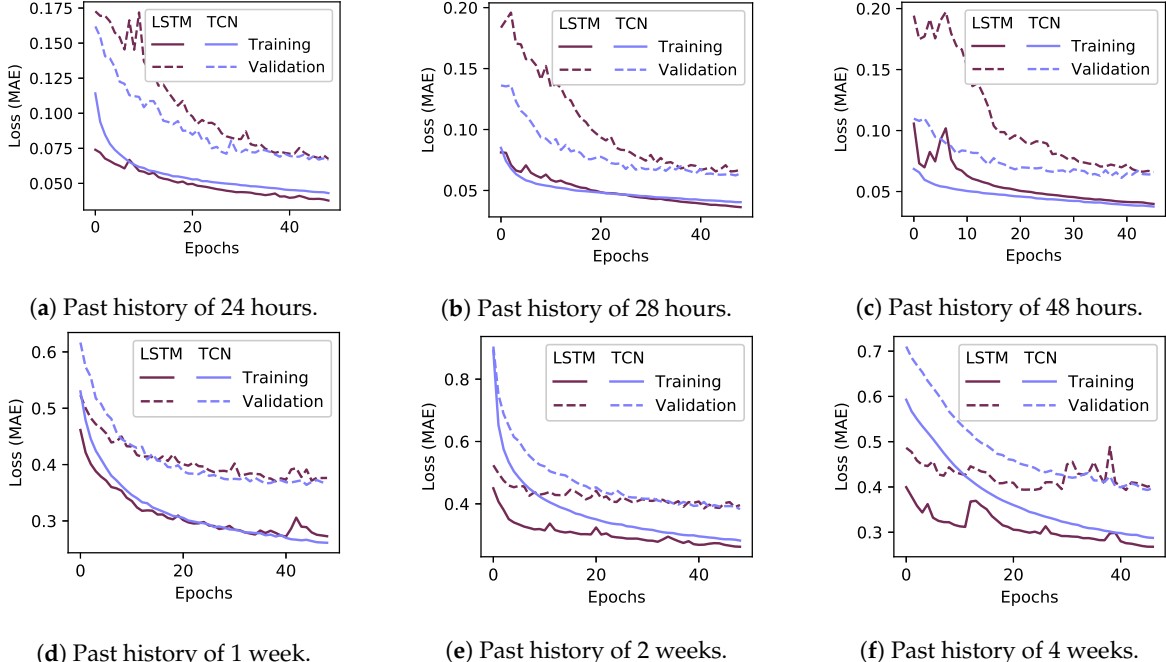

**Figure 8.** Evolution of the training and validation loss function of the best TCN and LSTM models for each possible value of past history. First row corresponds to the electric demand data (**a–c**), while the second row corresponds to the EV power consumption data (**d–f**).

### 3.2. Computation Time

The deep learning models proposed in this study have complex architectures that involve a high computational cost. Therefore, it is also essential to evaluate the models in terms of computational efficiency. Table 4 presents the training and prediction time for all the best TCNs and LSTMs that were presented in the previous subsection. It also reports the number of trainable parameters to further enhance the comparative study. For obtaining these computation times, the batch size has been fixed to 128 instances for all models in order to perform fair comparisons. As can be seen, TCNs have far more trainable parameters since their stacked blocks have many convolutional operations. This results in higher training time compared to the LSTM models. The differences in training times are slightly larger in the case of the electric demand data, considering that an epoch of this dataset involves many more instances than the EV data. However, with respect to prediction time, TCN models are always faster than recurrent networks. Regardless of the considerably higher number of parameters, TCNs are able to provide very quick forecasts once they are trained. As it was expected, parallel convolutions of TCNs can process long input sequences faster than the sequential processing of recurrent networks. This seems an essential advantage for real-time applications, in which predictions need to be obtained as soon as possible in order to make informed decisions.

**Table 4.** Computation time of the best TCN and LSTM models for each dataset. Prediction time corresponds to the time that a model takes to compute a prediction for one instance. Training time corresponds to the time that a model takes to run one training epoch.

**a** Computation time for electric demand data.

| Past History | | Model | | Trainable Paramereters | Prediction Time (ms) | Training Time (s) |
|---|---|---|---|---|---|---|
| Time-Steps | Hours | # | Type | | | |
| 144 | 24 | 1 | LSTM | 332,824 | 0.859 | **52.24** |
| | | 2 | TCN | 1,694,448 | **0.566** | 87.29 |
| 168 | 28 | 3 | LSTM | 332,824 | 0.968 | **58.56** |
| | | 4 | TCN | 1,694,448 | **0.590** | 99.44 |
| 288 | 48 | 5 | LSTM | 201,240 | 0.775 | **62.70** |
| | | 6 | TCN | 2,121,200 | **0.672** | 109.2 |

**b** Computation time for EV power consumption data.

| Past History | | Model | | Trainable Paramereters | Prediction Time (ms) | Training Time (s) |
|---|---|---|---|---|---|---|
| Time-Steps | Days | # | Type | | | |
| 168 | 7 | 7 | LSTM | 332,824 | 0.532 | **3.01** |
| | | 8 | TCN | 1,694,448 | **0.347** | 4.41 |
| 336 | 14 | 9 | LSTM | 332,824 | 1.020 | **6.76** |
| | | 10 | TCN | 1,694,448 | **0.836** | 10.49 |
| 672 | 28 | 11 | LSTM | 201,240 | 1.148 | **4.76** |
| | | 12 | TCN | 2,121,200 | **0.726** | 10.18 |

## 4. Conclusions

In this paper, we proposed a deep learning model based on temporal convolutional networks (TCN) to perform forecasting over two energy-related time series. The experimental study considered two real-world time series data from Spain: the national electric demand and the power demand at charging stations for electric vehicles. An extensive parameter search was conducted in order to obtain the best architecture configuration, testing more than 1200 different TCN models for both dataset. Furthermore, the performance of these convolutional networks was compared in terms of accuracy and efficiency with long short-term memory (LSTM) recurrent networks—that have so far been considered the state-of-the-art for forecasting tasks.

The results of the experimental study carried out showed that TCNs outperformed the forecasting accuracy of LSTM models for both datasets. The dilated causal convolutions used by TCNs were more effective at capturing temporal dependencies than the recurrent LSTM units. Furthermore, TCNs proved to be less sensitive to the parameter selection than LSTM models. Regardless of the chosen values, the convolutional approach provided a more reliable performance. Moreover, we also aimed to illustrate the importance of the size of the past history input window. Thanks to the use of residual connections, TCNs provided better results when using longer input sequences. In contrast, LSTM models were more accurate at encoding patterns when using smaller windows.

Regarding the computational efficiency, it was seen that TCN models have deeper architectures with many more trainable parameters. This implied that the training procedure of a TCN was slightly more costly. However, once TCNs were trained, they provided significantly faster predictions than recurrent networks due to the use of parallel convolutions to process the input sequences. In conclusion, our study demonstrated that TCNs are a very powerful alternative to LSTM networks. They can

provide more accurate predictions and are more suitable for real-time applications given their faster predicting speed.

Future efforts on this path will be focused on analysing the use of ensembles of TCN blocks with different receptive fields and using techniques such as evolutionary algorithms for the parameter search process. Another interesting future work could be the application of TCN networks in an online environment for real-time data streaming forecasting. Moreover, further research should also study the suitability of TCN networks for other problems like multivariable time series forecasting or time series classification.

**Author Contributions:** All authors made substantial contributions to conception and design of the study. P.L.-B. and M.C.-G. performed the experiments, analysed the data, and wrote the paper. J.M.L.-R. and J.C.R. guided the research and reviewed the manuscript. All authors have read and agreed to the published version of the manuscript.

**Funding:** This research has been funded by the Spanish Ministry of Economy and Competitiveness under the project TIN2017-88209-C2-2-R and by the Andalusian Regional Government under the projects: BIDASGRI: Big Data technologies for Smart Grids (US-1263341), Adaptive hybrid models to predict solar and wind renewable energy production (P18-RT-2778).

**Acknowledgments:** We are grateful to NVIDIA for their GPU Grant Program that has provided us the high-quality GPU devices for carrying out the study.

**Conflicts of Interest:** The authors declare no conflict of interest. The funders had no role in the design of the study; in the collection, analyses, or interpretation of data; in the writing of the manuscript, or in the decision to publish the results.

## Abbreviations

The following abbreviations are used in this manuscript:

| | |
|---|---|
| ANN | Artificial Neural Network |
| CNN | Convolutional Neural Network |
| DL | Deep Learning |
| EV | Electric vehicle |
| LSTM | Long Short-Term Memory Network |
| MAE | Mean Absolute Error |
| MIMO | Multi-Input Multi-Output |
| RNN | Recurrent Neural Network |
| SVM | Support Vector Machine |
| TCN | Temporal Convolutional Network |
| WAPE | Weighted Absolute Percentage Error |

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
