# Peer review of "Temporal Convolutional Networks Applied to Energy-Related Time Series Forecasting"

_applsci, doi:10.3390/app10072322_

Round 1

Reviewer 1 Report

The paper is well written and easy to follow. The only suggestion to improve the paper is to use a methodology to hyperparameter tunning instead of using an intensive search.

Reviewer 2 Report

Dear Authors,

The paper addresses a very current topic as it is electricity demand forecasting. TCN deep learning algortythms has been proposed and tested to forecast long term time-series. Results seem to be promising and the paper is acceptable with the exception of some remarks that I highly recommend that you accomplish:

1) line 196. Define here what is high accuracy: "The aim of this work is to build a TCN-based deep learning model that can achieve high 197 accuracy in energy demand forecasting tasks."

2) Forecasting accuracy is tested by box plots. This accuracy gathers information regarding how many and which error sources?

3) Is computation time reasonable in order to make real predictions ?

4) Which are the real possibilities of implementing this method ?

5) Have you estimated real costs of energy that this method can perform ? Put an example.

Thanks for your work

Round 2

Reviewer 2 Report

From my point of view, the paper ie ready.